# Direct observation of spin-layer locking by local Rashba effect in monolayer semiconducting PtSe$_2$ film

Wei Yao[1], Eryin Wang[1], Huaqing Huang[1], Ke Deng[1], Mingzhe Yan[1], Kenan Zhang[1], Koji Miyamoto[2], Taichi Okuda[2], Linfei Li[3], Yeliang Wang[3,4], Hongjun Gao[3,4], Chaoxing Liu[5], Wenhui Duan[1,4] & Shuyun Zhou[1,4]

The generally accepted view that spin polarization in non-magnetic solids is induced by the asymmetry of the global crystal space group has limited the search for spintronics materials mainly to non-centrosymmetric materials. In recent times it has been suggested that spin polarization originates fundamentally from local atomic site asymmetries and therefore centrosymmetric materials may exhibit previously overlooked spin polarizations. Here, by using spin- and angle-resolved photoemission spectroscopy, we report the observation of helical spin texture in monolayer, centrosymmetric and semiconducting PtSe$_2$ film without the characteristic spin splitting in conventional Rashba effect (R-1). First-principles calculations and effective analytical model analysis suggest local dipole induced Rashba effect (R-2) with spin-layer locking: opposite spins are degenerate in energy, while spatially separated in the top and bottom Se layers. These results not only enrich our understanding of the spin polarization physics but also may find applications in electrically tunable spintronics.

[1] State Key Laboratory of Low Dimensional Quantum Physics and Department of Physics, Tsinghua University, Beijing 100084, China. [2] Hiroshima Synchrotron Radiation Center (HSRC), Hiroshima University, 2-313 Kagamiyama, Higashi-Hiroshima 739-0046, Japan. [3] Institute of Physics & University of Chinese Academy of Sciences, Chinese Academy of Sciences, Beijing 100190, China. [4] Collaborative Innovation Center of Quantum Matter, Beijing, China. [5] Department of Physics, The Pennsylvania State University, University Park, State College, Pennsylvania 16802-6300, USA. Correspondence and requests for materials should be addressed to S.Z. (email: syzhou@mail.tsinghua.edu.cn).

The new insight that spin polarization in non-magnetic materials originates from relativistic spin–orbit coupling (SOC) by the local asymmetry (atomic site group)[1–4] rather than the global asymmetry (bulk space group) has led to two forms of hidden spin polarization in centrosymmetric materials, local Rashba R-2 (by site dipole field) and local Dresselhaus D-2 (by site inversion asymmetry) effect, which are distinguished from the conventional Rashba R-1 (by structural inversion asymmetry)[5–12] and Dresselhaus D-1 (by bulk inversion asymmetry)[13] effects previously discovered in non-centrosymmetric bulk materials or interfaces. The R-1 effect is induced by a potential gradient from the substrate or interface. Such structural inversion asymmetry lifts the spin degeneracy and leads to a net spin polarization with helical spin texture. Different from the R-1 effect, the R-2 effect is induced by local dipole fields and can exist in centrosymmetric materials. The compensated spins of opposite signs are degenerate in energy, while spatially locked to two real space sectors forming the inversion partners (for example, top and bottom layers) resulting in spin-layer locking. Compared with the R-1 effect with a large internal electric field, which is difficult to be controlled by external methods, the R-2 effect may have advantages for electrically tunable spintronics devices due to the easy manipulation of spins via the application of an external electric field[14,15]. A critical step towards this goal is to realize stable semiconducting thin films with large spin polarizations. Layered transition metal dichalcogenides[16–19] are promising candidates for realizing hidden spin polarizations due to the strong SOC and large site dipole field. Although hidden spin polarization by D-2 effect has been reported in the bulk crystal of $WSe_2$ (ref. 3), the spin-layer locking induced by the R-2 effect in thin semiconducting films has not been realized experimentally yet.

Here, by combining spin- and angle-resolved photoemission spectroscopic (spin-ARPES) measurements with three-dimensional spin analysis and theoretical calculations, we report the hidden helical spin texture and spin-layer locking induced by R-2 effect in a monolayer semiconducting $PtSe_2$ film.

## Results

**Crystal and electronic structures.** Bulk $PtSe_2$ (ref. 20) is similar to the Dirac semimetal $PtTe_2$ (ref. 21), possessing centrosymmetric space group $P\bar{3}m1$ for the global structure and polar point groups $C_{3v}$ and $D_{3d}$ for the Se and Pt sites, respectively. Monolayer $PtSe_2$ contains one Pt layer sandwiched between two Se layers, forming trigonal structure when projected onto the (001) plane (Fig. 1a). First-principles calculations (Fig. 1b) show that the top three valence bands (labelled by $\alpha$, $\beta$ and $\gamma$) are mostly contributed by the $p$ orbitals of Se and the fourth valence band (labelled by $\delta$) is mainly contributed by the $d$ orbitals of Pt. As monolayer $PtSe_2$ has both inversion and time-reversal symmetries, all these bands are expected to be doubly degenerate without any net spin polarization.

Figure 1c shows the low-energy electron diffraction pattern of the high-quality monolayer $PtSe_2$ thin film grown on Pt(111) substrate[22]. The semiconducting property of monolayer $PtSe_2$ has been reported in previous work[22]. ARPES data measured along two high-symmetry directions M-Γ-M (Fig. 1d) and K-Γ-K (Fig. 1e), both show four valence bands (Fig. 1f) with similar dispersions, suggesting that the electronic structure is overall rather isotropic. Correspondingly, the constant energy maps (Fig. 1h) show a circular shape for all the bands around the Γ point and hexagonal warping is observed mainly at larger momenta. Analysis from energy distribution curve shows that the four valence bands are separated from each other at the

Γ point (Fig. 1g). Although the measured dispersions (Fig. 1d,e) show a shift of energy by ≈ 1.14 eV compared with the calculated dispersions (Fig. 1b) due to charge transfer from the substrate, the good agreement between the measured and calculated dispersions suggests that the hybridization with the Pt(111) bands is weak. Further calculations combining the Pt(111) substrate show that the influence of the substrate on the electronic structure of $PtSe_2$ film is negligible, except a small band splitting (~15 meV), which is equivalent to the effect of a small electric field (~$0.1\,V \cdot \text{Å}^{-1}$; see details in Supplementary Fig. 1a–c and Supplementary Note 1). In fact, the splitting is too small to be observed in experiments; thus, the sample measured here can be approximated as quasi-freestanding monolayer $PtSe_2$.

**Spin polarization along M-Γ-M direction.** Although both the measured and calculated dispersions show negligible spin splitting, spin-ARPES measurements reveal surprisingly large spin polarizations. Figure 2 shows three-dimensional spin analysis for the measurements along the M-Γ-M direction. A large spin contrast is observed along the tangential direction ($\theta$) for $\beta$, $\gamma$ and $\delta$ bands at emission angle of 7.5° (dashed line in Fig. 2a) with up to 50% polarization (Fig. 2c), whereas negligible spin contrast is observed along the radial ($r$) and out-of-plane ($\perp$) directions (Fig. 2b,d). The spin directions are illustrated by blue crosses (into the plane) and red dots (out of the plane) in Fig. 2a. We extend the in-plane tangential spin analysis to other momentum positions along the M-Γ-M direction. Figure 2e shows the intensity distribution at different emission angles for spin-up and spin-down states along the tangential direction and the spin polarization is shown in Fig. 2f. When approaching the Γ point from the M point, the degree of spin polarization decreases and it becomes negligible at the Γ point. On the other side of the Γ point, the spin polarization is reversed. The observation of spin polarization along the in-plane tangential direction with opposite signs on the two sides of the Γ point is consistent with helical spin texture.

**Spin polarization along K-Γ-K direction.** To confirm the helical spin texture, we show in Fig. 3 the spin analysis for data measured along the other high-symmetry direction K-Γ-K. A large spin polarization is also observed for the tangential component (Fig. 3b,c), supporting that the spin texture is helical. The $\alpha$-band shows negligible spin polarization near the Γ point. When moving to a larger emission angle along the Γ-K direction, the $\alpha$-band is separated from other bands and we can resolve its spin polarization more easily here. In Fig. 3d, spin polarization is observed in the $\alpha$-band and its direction is the same as the $\beta$-band. Considering that the electronic structure is rather isotropic, the observed large spin polarization along the tangential direction for both Γ-K and Γ-M directions suggests that the spin polarization has an overall helical texture. Different from the Γ-M direction, a small yet detectable spin polarization of ≈ 5% is observed in the radial component for $\beta$- and $\gamma$-bands with opposite spin directions (Fig. 3f,g) and its polarization does not reverse at the opposite sides about the Γ point. This suggests that the spin structures near the K and K′ points in Brillouin zone are not equivalent. This could be related to the defects in sample or other underlying mechanism such as final-state effect. Combining all spin-ARPES measurements discussed above, monolayer $PtSe_2$ on Pt(111) shows overall helical spin textures with their helicities summarized in the schematic drawing in Fig. 3h.

## Discussion

Although it is tempting to attribute the observed helical spin texture to R-1 Rashba effect induced by the structural asymmetry

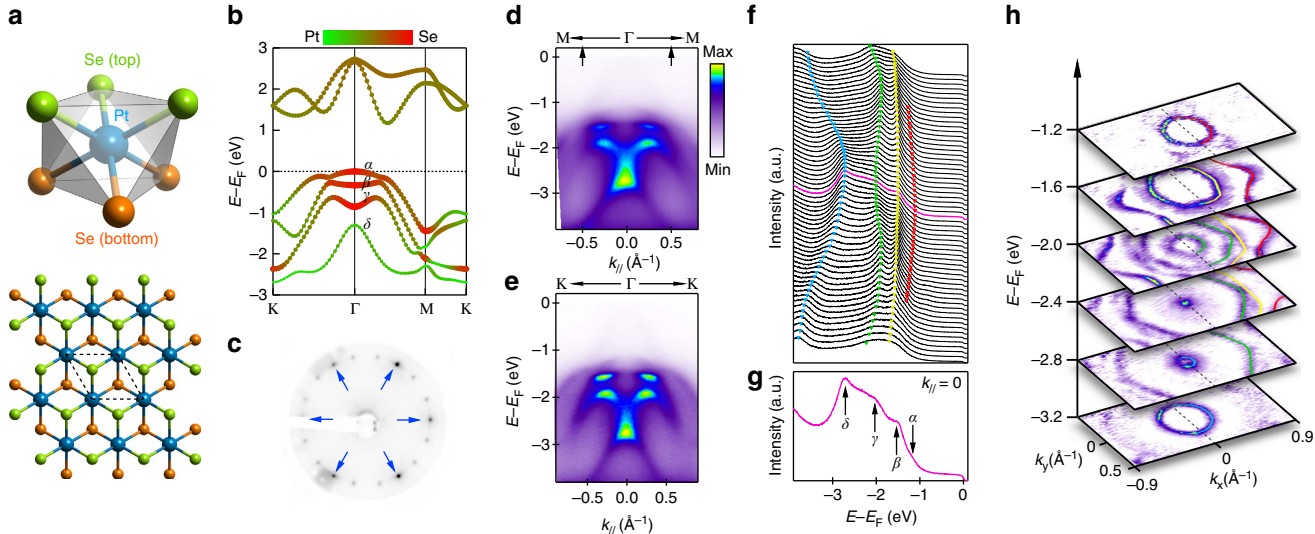

**Figure 1 | Crystal structure and electronic structure of monolayer PtSe₂.** (**a**) Crystal structure of monolayer PtSe₂ (top: unit cell, bottom: top view). (**b**) Band dispersions along the K-Γ-M-K direction from first-principles calculations. The colour and line width distinguish the contribution from Pt and Se (Red for Se and green for Pt). (**c**) Low-energy electron diffraction (LEED) pattern of as-grown monolayer PtSe₂ on Pt(111). The peaks from PtSe₂ are pointed by blue arrows. Additional diffraction spots rotated from the PtSe₂ spots are much weaker and at different azimuthal angles and the measured band dispersions in **h** do not show contributions from them, which are expected at different azimuthal angles. (**d**) ARPES data measured along M-Γ-M direction. (**e**) ARPES data measured along the K-Γ-K direction. (**f**) Energy distribution curves (EDCs) for data shown in **d** for momentum range between the arrows. (**g**) The EDC at Γ point, from which we can resolve the four bands. (**h**) Constant energy maps at selected energies. The contours of the top four bands at positive $k_x$ are highlighted by coloured lines.

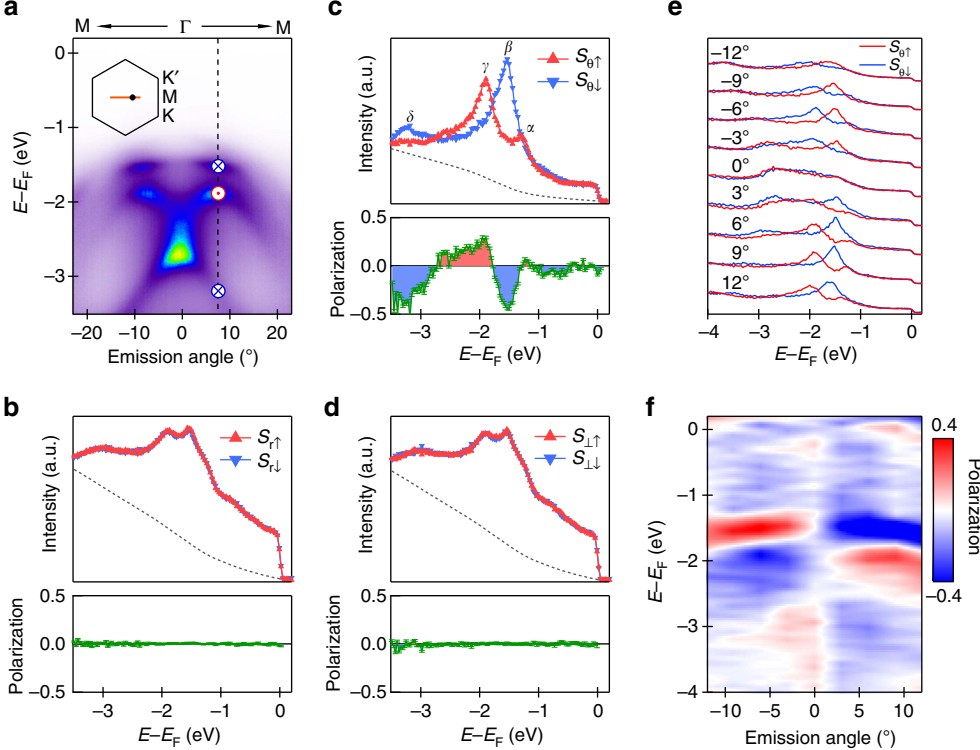

**Figure 2 | Spin polarization for the bands along the M-Γ-M direction.** (**a**) ARPES data measured at different emission angles. The inset shows the position of this cut in the Brillouin zone and the dot marks the position for energy distribution curves (EDCs) shown in **b**–**d**. The red dots and blue crosses stand for the spin directions. (**b**–**d**) Spin-resolved EDCs of three spin components (in-plane radial, tangential directions and out-of-plane direction) at emission angle of 7.5° (along the dashed line in **a**). The EDCs have been corrected by Shirley background (broken lines). Lower panels are extracted spin polarizations. The error bars indicate the statistical fluctuation in determining the spin polarizations (proportional to $\frac{1}{\sqrt{N}}$, where $N$ is the photoemission intensity). (**e**) EDCs of spin-up and spin-down states at different emission angles. (**f**) Spin polarization at different emission angles. Red indicates spin-up states and blue indicates spin-down states along tangential direction.

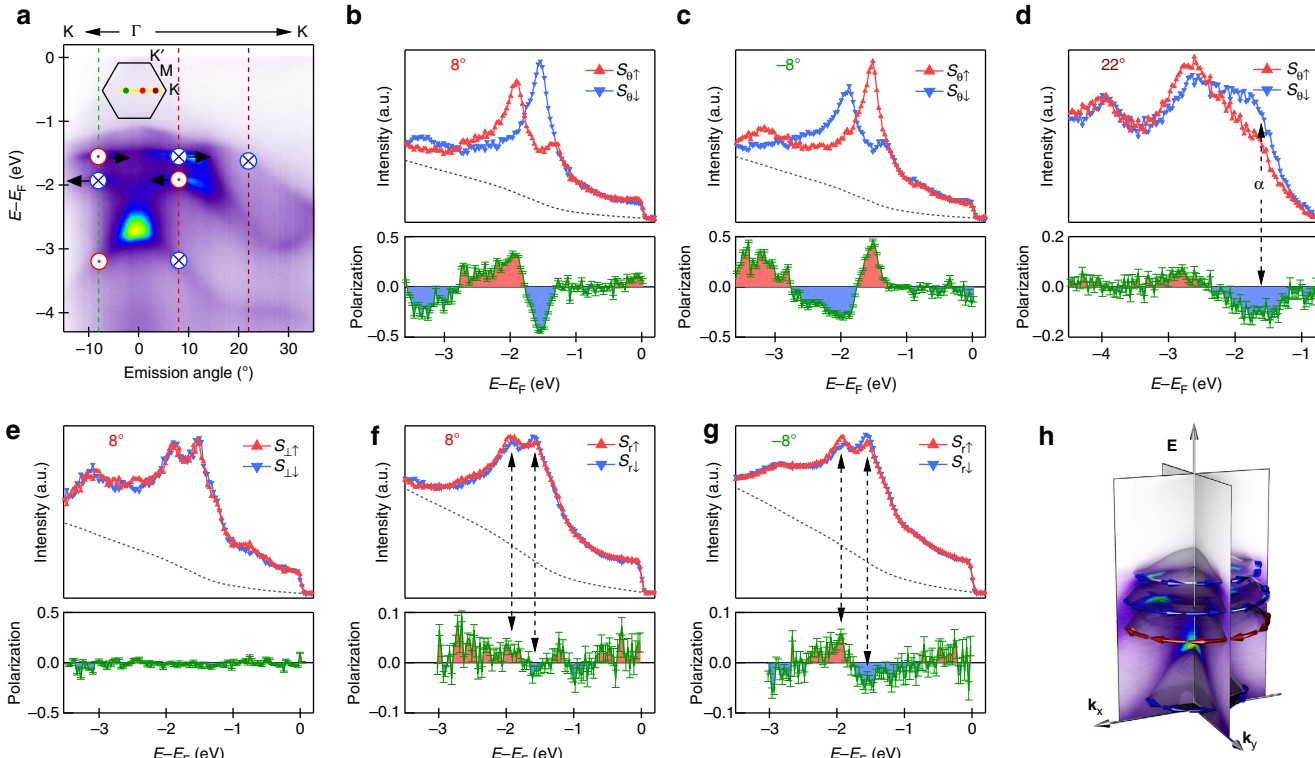

**Figure 3 | Spin texture measured along the K-Γ-K direction and the overall spin texture.** (**a**) ARPES data measured at different emission angles. (**b,c**) Spin-resolved energy distribution curves (EDCs) for the in-plane tangential direction at emission angles of 8° and −8°, respectively. The error bars indicate the statistical fluctuation in determining the spin polarizations (proportional to $\frac{1}{\sqrt{N}}$, where $N$ is the photoemission intensity). (**d**) Spin-resolved EDCs for the in-plane tangential direction at emission angle of 22°. (**e**) Spin-resolved EDCs along the out-of-plane direction at emission angle of 8°. (**f,g**) Spin-resolved EDCs for the radial direction at emission angles of 8° and −8°, respectively. (**h**) Overview of the spin texture of monolayer PtSe$_2$ from spin-ARPES measurements.

from the Pt substrate, further analysis, however, shows that this mechanism cannot explain our observations. For energy bands with the R-1 effect, there are two key features as follows: (1) two spin states are split at finite momenta and become degenerate at the Γ point due to the time-reversal symmetry (Fig. 4a) and (2) opposite helical spin textures with the same degree of polarization should exist for two spin-split bands so that they cancel with each other, leading to zero net spin polarization at one momentum. We find that the spin polarization observed for the four bands ($\alpha$, $\beta$, $\gamma$ and $\delta$) in experiments cannot satisfy these two features. One possibility is to assign $\alpha$- and $\beta$-bands as one pair of spin-split bands, and the $\gamma$- and $\delta$-bands as another pair. This assignment is clearly incorrect because of the following two reasons: (1) a large energy gap is found between the $\alpha$ ($\gamma$)- and $\beta$ ($\delta$)-bands at the Γ point (Fig. 1g). This means a strong time-reversal symmetry breaking in our system and is impossible for a non-magnetic system. (2) Spin texture is almost vanishing for the $\alpha$-band at a small momentum, whereas it shows a large value for the $\beta$-band, inconsistent with the requirement (2) of the R-1 effect. This argument, in particular the requirement (1) of the R-1 effect, suggests that even if the R-1 effect exists in our system due to the Pt substrate, the pair of spin-split bands only could be nearly degenerate (consistent with the band calculation in Fig. 1b or Supplementary Fig. 1a); hence, for any band of $\alpha$, $\beta$, $\gamma$ and $\delta$ observed in the ARPES measurements, it should already include the contribution from both spin-split states. However, the net spin polarization should vanish according to the requirement (2). Thus, we conclude that the simple R-1 effect cannot explain the observed helical spin texture.

This leads to the speculation that the helical spin texture observed is induced by R-2 Rashba effect (Fig. 4b). Next, we will demonstrate that the R-2 effect and the spin-layer locking indeed can provide a natural explanation of the spin textures based on some simple assumptions.

To verify our speculation, we calculate the projected spin polarization onto the top and bottom Se layers. Helical spin textures indeed emerge in each Se layer with opposite helicities, which result from the R-2 effect. The obtained spin textures on the top Se layer in Fig. 5a reproduce qualitative features observed in spin-ARPES measurements for all four bands. For example, the $\alpha$- and $\beta$-bands indeed have the same spin polarization at large momentum, whereas those of $\gamma$- and $\delta$-bands are opposite. For the $\alpha$-band, the spin polarization is negligible compared with that of the $\beta$-band close to the Γ point, which is also similar to the observed spin polarization. To explain the spin ARPES measurements based on our calculations, we make the assumption that spin polarization from the top Se layer has a stronger contribution compared with that from the bottom Se layer due to the limited escape length of photoelectrons in the ARPES measurement. The good agreement between experimental results and the calculations provides strong support for the R-2 effect and indicates a new spin-layer locking physics (locking in the atomic sub-layer) (Fig. 5c). One can easily note that this is different from that in the 2H structure (for example, bulk WSe$_2$ reported in ref. 3), where the spin-layer locking happens in the entire single-unit layer and is induced by the D-2 effect. Further calculations for the effect of the substrate on the spin texture are shown in Supplementary Fig. 1d and the dominant

role of the local dipole field on the helical spin texture is confirmed. The existence of the new spin-layer locking can be understood as a general consequence of the common sandwich

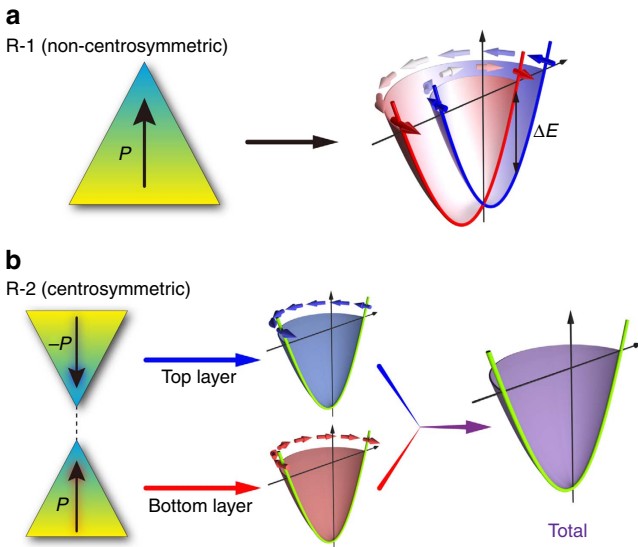

**Figure 4 | Comparison between conventional Rashba (R-1) and local Rashba (R-2) effects.** (**a**) R-1 Rashba effect in non-centrosymmetric materials. The inversion symmetry is broken and there is a net dipole field. The right panel shows schematically the typical spin-resolved band structure induced by R-1 Rashba effect with a spin splitting in energy. (**b**) R-2 Rashba effect in centrosymmetric materials. The inversion symmetry is preserved and there is a local site dipole field in spite of the vanishing total dipole field. The right panels illustrate the schematic spin texture induced by R-2 Rashba effect. The two kinds of spin texture with opposite helicities are spatially separated in different layers, resulting in overall zero spin polarization.

type of crystal structures in all these materials, in which local electric fields are expected to point from the two outer layers to the central layer. Although such a spin-layer locking has been predicted in other layered materials[1,15,23,24], our work experimentally realizes a stable semiconducting thin film with large spin polarization induced by the R-2 effect.

More theoretical understanding can be obtained by analysing the orbital natures of these four bands near the $\Gamma$ point (the atomic limit, $\mathbf{k} \sim 0$, as shown in Fig. 5b) and this also confirms the spin texture discussed above. The bands near the Fermi energy are dominated by the $p$ orbitals of Se atoms and the $d$ orbitals of Pt atoms. The strong anisotropy introduces a strong energy splitting between the $p_{x,y}$ orbitals and the $p_z$ orbital of Se atoms, pushing Se $p_z$ orbitals to lower energy. As a result, the conduction and valence bands around the band gap mainly consist of the $p_{x,y}$ orbitals of Se atoms. The hybridization of the Se $p_{x,y}$ orbitals between the top and bottom Se layers can be mediated by the central Pt layer and leads to a band gap opening. After taking into account the SOC, we find that the $\alpha$- and $\beta$-bands correspond to the bonding states of Se $p_{x,y}$ orbitals with total $z$-direction angular momentum $\pm\frac{3}{2}$ and $\pm\frac{1}{2}$, respectively. We explicitly show the atomic orbital form of the basis wave function and construct the corresponding low-energy effective Hamiltonian for the $\alpha$- and $\beta$-bands based on the symmetry principle in the Supplementary Note 3. The effective Hamiltonian clearly shows layer-dependent spin textures for the $\beta$-bands. In addition, it is found that the vanishing spin texture of the $\alpha$-band at a small momentum results from the $\pm\frac{3}{2}$ angular momentum in contrast to the $\pm\frac{1}{2}$ angular momentum of the $\beta$-band.

In summary, we report the experimental realization of unconventional R-2 Rashba effect and intrinsic spin-layer locking in a centrosymmetric monolayer $PtSe_2$ thin film. Considering that the monolayer $PtSe_2$ sample is very stable, only monolayer thick, and semiconducting, growth or transfer of such thin film onto insulating substrates may provide exciting opportunities to realize electrically controllable spintronics devices.

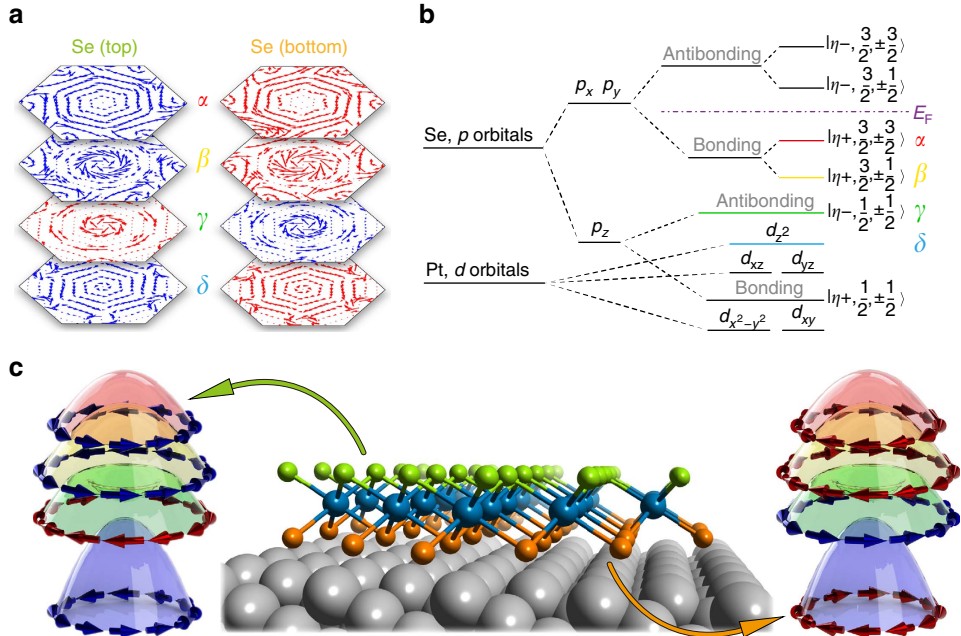

**Figure 5 | The unconventional spin-layer locking by R-2 effect proposed for monolayer PtSe₂.** (**a**) Spin texture of the two Se layers from first-principles calculation for the four bands, respectively. The arrow colours represent the spin helicities: the blue arrows are counterclockwise and the red arrows are clockwise. The arrow size indicates the strength of spin polarization. (**b**) The diagram of Se $p$ orbitals and Pt $d$ orbitals, which dominate the spin texture of bands near Fermi level. (**c**) The schematic diagram for the new spin-locking mechanism of monolayer PtSe₂ thin film on Pt(111).

## Methods

**Experiments.** The PtSe$_2$ thin film was grown by direct selenization of Pt(111)[22]. The growth stops when the substrate is covered by one monolayer of PtSe$_2$. Spin-ARPES measurements were performed at ESPRESSO endstation[25,26] of Hiroshima Synchrotron Radiation Center. The normal and spin ARPES measurements were both taken at 20 K, using photon energies of 21.2 eV (UV lamp) and 20 eV (Synchrotron radiation). These two light sources give the same spin structures (see Supplementary Fig. 2 and Supplementary Note 2). Spin polarizations are calculated by $P = A/S_{eff}$, where $A = (I_+ - I_-)/(I_+ + I_-)$ is the intensity asymmetry for different magnetization directions of the detector target and $S_{eff}$ is the effective Sherman function for the spin detector. For radial and out-of-plane components the effective Sherman function is 0.3, and for tangential component the effective Sherman function is 0.235.

**Calculations.** The first-principles calculations are performed using the density functional theory as implemented in the Vienna *ab initio* simulation package[27] with the projector augmented-wave method. Perdew–Burke–Ernzerhof parametrization of the generalized gradient approximation is used for the exchange-correlation potential[28]. We adopt a default plane-wave energy cutoff and the Brillouin zone is sampled by a $\Gamma$-centered $6 \times 6 \times 1$ $k$-point mesh. The monolayer structure of PtSe$_2$ is modelled with a vacuum region more than 15 Å thick, to eliminate the spurious interaction between neighbouring layers. A perpendicular electric field of $0.1 \, \text{V} \cdot \text{Å}^{-1}$ was applied to simulate the effect of the substrate. SOC is included in all electronic structure calculations.

**Data availability.** All relevant data within the paper and its Supplementary Information are available from the corresponding author upon reasonable request.

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

## Acknowledgements

This work was supported by the Ministry of Science and Technology of China (Grant Numbers. 2015CB921001, 2016YFA0301004 and 2016YFA0301001) and the National Natural Science Foundation of China (Grant Numbers 11274191, 11334006 and 11427903). C.-X.L. acknowledges the support from Office of Naval Research (Grant Number N00014-15-1-2675). Spin-ARPES experiments at Hiroshima Synchrotron Radiation Center have been performed under the proposal number 14-A-15 and 16AG058.

## Author contributions

S.Z. designed the research project. W.Y. and K.D. prepared the samples with coordination by L.F.L., Y.L.W. and H.J.G. W.Y., E.W., K.D., M.Y., K.Z. and S.Z. performed the spin-ARPES measurements and data analysis with assistance from K.M. and T.O. H.H. and W.D. performed the first-principles calculations and C.L. worked out the effective analytical model. W.Y., C.L. and S.Z. wrote the manuscript and all authors commented on the manuscript.

## Additional information

**Publisher's note**: 

