## [Peer Review File · Nature Communications]

Reviewer #1 (Remarks to the Author):

I recommend this work for publication in Nature Comms. I have evaluated it following the suggested criteria from the journal. I have one minor suggestion for improvement.

A: Summary of the key results

The manuscript investigates the unusual spin texture of the centrosymmetric semiconductor PtSe₂. The unusual spin texture is measured and understood in terms of local (rather than global) symmetry breaking.

B: Originality and interest: if not novel, please give references

I think there is a high degree of originality and interest. There is a clear connection to the related works of Refs 2-5. The authors are aware of the relevant work and their own work is a thorough and interesting development in this area.

C: Data & methodology: validity of approach, quality of data, quality of presentation

Very good. I especially appreciate that the language is clear and concise.

D: Appropriate use of statistics and treatment of uncertainties

Very good. The spin-ARPES data is of world-class quality.

E: Conclusions: robustness, validity, reliability

Very good. Evidence and conclusions are clear and logical to follow.

F: Suggested improvements: experiments, data for possible revision

I would like to see a brief discussion of light polarisation. How can the authors be sure that their measured spin is an initial-state effect and not due to symmetry-breaking in the photoemission process? What is the polarisation of the light, relative to the sample, and what are the implications of this? (on a related note: it would be nice to see a photon-energy-dependence of the measured spin, but I appreciate that this is an impractical request. If the authors are able to add data at additional photon energies to the suppl. mat. then this would be nice - but not really necessary)

G: References: appropriate credit to previous work?

Very good.

H: Clarity and context: lucidity of abstract/summary, appropriateness of abstract, introduction and conclusions

Very good.

Reviewer #2 (Remarks to the Author):

The manuscript by Yao et al. reports about a joined experimental and theoretical work on spin polarisation properties of PtSe₂ monolayer on Pt(111). The subject of the study might be interesting for a broad community in condensed matter physics. Both experimental measurements and theoretical calculations have been made using reliable methods and the results have been correctly interpreted. In general, the manuscript is very well written and is interesting to read. However, I've several questions, which should be properly clarified by the authors before the manuscript can be recommended for publication.

1. What is a real band gap in PtSe₂ monolayer? Is this quantity accessible from experiment? The LDA calculations report the band gap of 1.2 eV, the GWA - 2.1 eV. How is the band gap size is

important for the interpretation provided in the work?

2. The calculations have been made for a single PtSe₂ monolayer in vacuum but not on Pt(111). The effect of the substrate was simulated by an applied electric field of 0.1 V/Å. The model should be justified. At least, the authors should show how the results depend on the electric field value. A more reliable way is to take into account the Pt(111) substrate. Why was it not done?

3. Fig. 4(h) shows the spin texture only schematically. This figure does not provide any clear information about the spin texture in PtSe₂. It would be more useful to show the results of spin-polarised ARPES as it was done for conventional ARPES (Fig. 2(h)). A such result can be directly compared with the theory.

4. Fig. 5(a) is not clear for me. First, what is color schema in this figure? How can I see that the alpha and beta bands have the same spin polarisation at large momentum but the gamma and delta opposite? Opposite to what? The spin texture is a quantity which depends on energy. At which energies are the results shown on Fig. 5(a)?

Reviewer #3 (Remarks to the Author):

The theoretical work by Zhang and co-workers [Ref. 2] opened a door to search for spin-polarization effects in centrosymmetric materials with potential for tunable spintronic applications. This paper presents a combined spin- and angle-resolved photoemission spectroscopy study of monolayer PtSe₂ thin film. With the observation of a helical spin texture without characteristic spin splitting in conventional Rashba effect, the authors claim a local dipole induced Rashba (R-2) effect with spin-layer locking. Such a finding would be the first experimental realization of unconventional R-2 Rashba effect predicted by the theory, hence is an important step towards designing future spintronics devices.

There are a number of issues to clarify. First of all, for the monolayer PtSe₂, the bond length of Pt-Se can be slightly different for the top Se layer compared with the bottom Se layer, either due to the charge transfer at the interface or the surface termination of the top Se layer. Such an asymmetry in bond length, if presents, would break the inversion symmetry of the PtSe₂ monolayer. One may wonder whether the observed spin polarization could be a trivial result of lacking the inversion asymmetry in the monolayer.

Next, the observed spin polarization is attributed to limited escape depth of photoelectrons that results in imbalanced contributions from the top and bottom Se layer. It would be enlightening to estimate the degree of the spin polarization based on the difference in probing depth. In this regard, it could also be helpful to repeat spin-ARPES measurements at different photon energy with different probing depth to confirm the origin of the observed spin polarization.

Last, it was proposed in the paper that the existence of the new spin-layer locking as a general consequence of the common sandwich type of crystal structures in layered TMDs. If so, one may wonder whether similar R-2 effect should also be observed in bulk crystal of WSe₂ as similar argument of photoelectron depth should also be applicable. In fact, previous spin-ARPES experiments [Ref. 4] reported hidden spin polarization due to D-2 effect. It would be helpful for the authors to comment in the paper the difference between the PtSe₂ monolayer and bulk crystal of WSe₂.

In summary, this paper reports the first direct observation of R-2 Rashba effect in monolayer PtSe₂, which would make a timely contribution to the field in searching for spin polarization in centrosymmetric materials. The data presented in the paper is of high quality and the data presentation is well organized. I would recommend this paper for publication in Nature Communications, provided that the authors can satisfactorily address the above issues.

Reviewer #1 (Remarks to the Author):

I recommend this work for publication in Nature Comms. I have evaluated it following the suggested criteria from the journal. I have one minor suggestion from improvement.

Reply: We thank the reviewer #1 for his/her recommendation for publication.

1. Summary of the key results

The manuscript investigates the unusual spin texture of the centrosymmetric semiconductor PtSe₂. The unusual spin texture is measured and understood in terms of local (rather than global) symmetry breaking.

2. Originality and interest: if not novel, please give references

I think there is a high degree of originality and interest. There is a clear connection to the related works of Refs 2-5. The authors are aware of the relevant work and their own work is a thorough and interesting development in this area.

3. Data & methodology: validity of approach, quality of data, quality of presentation

Very good. I especially appreciate that the language is clear and concise.

4. Appropriate use of statistics and treatment of uncertainties

Very good. The spin-ARPES data is of world-class quality.

5. Conclusions: robustness, validity, reliability

Very good. Evidence and conclusions are clear and logical to follow.

6. Suggested improvements: experiments, data for possible revision

I would like to see a brief discussion of light polarization. How can the authors be sure that their measured spin is an initial-state effect and not due to symmetry-breaking in the photoemission process? What is the polarization of the light, relative to the sample, and what are the implications of this? (on a related note: it would be nice to see a photon-energy-dependence of the measured spin, but I appreciate that this is an impractical request. If the authors are able to add data at additional photon energies to the suppl. mat. then this would be nice - but not really necessary)

Reply: In our work, we use two kinds of light sources to measure the spin polarization, unpolarized UV lamp at 21.2 eV and synchrotron radiation with linear polarization (see the methods section in main text). Both of these two measurements give the similar results (see the figures below), confirming that the measured spin polarization is not related to the polarization of the incident light. We have included this part in the revised supplementary information. The current results are sufficient to reveal the R-2 effect and the photon-energy-dependent measurements are in our future plan.

Fig. R1: The left panel shows the spin-resolved EDC at 6° emission angle along Γ -M direction by UV lamp. The right panel shows the spin-resolved EDC at 8° emission angle by synchrotron radiation. The similar results indicate that the spin polarization is not related to the polarization of light.

7. References: appropriate credit to previous work?

Very good.

8. Clarity and context: lucidity of abstract/summary, appropriateness of abstract, introduction and conclusions

Very good.

Reply: We appreciate the reviewer's positive comments on our work in the aspects above.

Reviewer #2 (Remarks to the Author):

The manuscript by Yao et al. reports about a joined experimental and theoretical work on spin polarization properties of PtSe₂ monolayer on Pt(111). The subject of the study might be interesting for a broad community in condensed matter physics. Both experimental measurements and theoretical calculations have been made using reliable methods and the results have been correctly interpreted. In general, the manuscript is very well written and is interesting to read. However, I've several questions, which should be properly clarified by the authors before the manuscript can be recommended for publication.

Reply: We thank the reviewer #2 for appreciating the scientific merits of our work.

1. What is a real band gap in PtSe₂ monolayer? Is this quantity accessible from experiment? The LDA calculations report the band gap of 1.2 eV, the GWA - 2.1 eV. How is the band gap size important for the interpretation provided in the work?

Reply: Since ARPES can only measure the occupied states, we are not able to access the energy gap in ARPES measurements. The energy gap has little relevance since we only concern the spin texture of valence bands in this work. However, it might be important for spintronics devices application in future, because the gap in PtSe₂ film is strongly

dependent on the layer number (see reference #21 in main text).

2. The calculations have been made for a single PtSe₂ monolayer in vacuum but not on Pt(111). The effect of the substrate was simulated by an applied electric field of 0.1 V/Å. The model should be justified. At least, the authors should show how the results depend on the electric field value. A more reliable way is to take into account the Pt(111) substrate. Why was it not done?

Reply: The justification has already been shown in the section I of the supplementary information with calculated results of PtSe₂ on Pt(111) substrate and we have added a summary paragraph in the main text. “Further calculations combining Pt(111) show that the influence of the substrate on the electronic structure of PtSe₂ is negligible except a small band splitting (~ 15 meV), which is equivalent to the effect of a small electric field (~ 0.1 V/Å) (see Fig.S1 in Supplementary Information).”

More detailed explanation is in Section I of Supplementary information.

“Figure S1(a) shows the calculated band structure of monolayer PtSe₂ on Pt(111) substrate, using an experimental value for the separation of 4.5 Å between the PtSe₂ and the substrate [1]. To extract the energy bands of monolayer PtSe₂, we project these bands onto Se atoms and show them as red dots. The projected band structure still retains the band structure of free-standing PtSe₂ (Fig. 2(b)), suggesting that the hybridization between Pt(111) and the sample is negligible. More detailed analysis (Fig. S1(b)) shows a small spin splitting of 15 meV in the β band of PtSe₂. Therefore, the effect of the substrate is mainly to induce a charge transfer and an electric field by a potential gradient. We further perform the calculation by applying a small external electric field on monolayer PtSe₂ to simulate the impact from substrate. Under an applied electric field of 0.1 V/Å, the β band opens a SOC gap (Fig. S1(c)) with a similar value to that induced by the substrate, hence such an electric field could mimic the effect from substrates. Under this condition, we analyze the spin texture following the same procedure in the main text. We find that spin textures in each layer are almost identical between the case of the free-standing film and that with an external electric field.”

3. Fig. 4(h) shows the spin texture only schematically. This figure does not provide any clear information about the spin texture in PtSe₂. It would be more useful to show the results of spin-polarized ARPES as it was done for conventional ARPES (Fig. 2(h)). A such result can be directly compared with the theory.

Reply: The two ways both have their own advantage and disadvantage. The constant energy map style (as Fig. 2(h)) can compare the spin texture of different bands at the same energy level, but it is not convenient to compare the spin polarization of different bands at the same momentum. The style of Fig. 4(h) balances the two aspects. On the other hand, the style of Fig. 4(h) is better for those who are not familiar with ARPES to understand our results.

4. Fig. 5(a) is not clear for me. First, what is color schema in this figure? How can I see that the alpha and beta bands have the same spin polarization at large momentum but the gamma and delta opposite? Opposite to what? The spin texture is a quantity which depends on energy. At which energies are the results shown on Fig. 5(a)?

Reply: The color of spin texture indicates its helicity and the small arrow points to the spin direction at each momentum position. For example, the red color means the spin rotates clockwise, and the blue color means the spin rotates counterclockwise. We have clarified this in revised manuscript. Figure 3(e, f) shows while the spin direction depends on the momentum (left or right side of the Gamma point), for a single band on one side of the Gamma point, the spin direction is the same.

Figure 5(a) is the projection of the spins into the in-plane momenta. Since every spin corresponds to a Bloch state, so the spin at different momentum has different energy according to the band dispersion.

Reviewer #3 (Remarks to the Author):

The theoretical work by Zhang and co-workers [Ref. 2] opened a door to search for spin-polarization effects in centrosymmetric materials with potential for tunable spintronic applications. This paper presents a combined spin- and angle-resolved photoemission spectroscopy study of monolayer PtSe₂ thin film. With the observation of a helical spin texture without characteristic spin splitting in conventional Rashba effect, the authors claim a local dipole induced Rashba (R-2) effect with spin-layer locking. Such a finding would be the first experimental realization of unconventional R-2 Rashba effect predicted by the theory, hence is an important step towards designing future spintronics devices.

Reply: We thank the reviewer #3 for considering our work as an important step towards designing future spintronics devices.

There are a number of issues to clarify. First of all, for the monolayer PtSe₂, the bond length of Pt-Se can be slightly different for the top Se layer compared with the bottom Se layer, either due to the charge transfer at the interface or the surface termination of the top Se layer. Such an asymmetry in bond length, if presents, would break the inversion symmetry of the PtSe₂ monolayer. One may wonder whether the observed spin polarization could be a trivial result of lacking the inversion asymmetry in the monolayer.

Reply: Such a kind of symmetry-breaking is equivalent to the conventional Rashba effect (R-1 effect), because it has broken the global inversion symmetry, but as we argued in the main text, the results of R-1 effect is different from the R-2 effect (existence of band splitting, band degeneration at Gamma point etc.), and we have ruled out the possibility of R-1 effect in our sample.

Next, the observed spin polarization is attributed to limited escape depth of photoelectrons that results in imbalanced contributions from the top and bottom Se layer. It would be enlightening to estimate the degree of the spin polarization based on the difference in probing depth. In this regard, it could also be helpful to repeat spin-ARPES measurements at different photon energy with different probing depth to confirm the origin of the observed spin polarization.

Reply: Actually, we can do a simple calculation to estimate the degree of spin polarization. First, we assume that the escape depth is around 5 Å according to the universal curve of photoelectron escape depth. Next we know that the separation between the two Se layers is about 2.6 Å. Then we can estimate that the ratio between the electron intensity from the two Se layers is about $\exp(-2.6/5) \approx 0.6$. Thus the estimated spin polarization is about $(1-0.6)/(1+0.6)=0.25$ which is a reasonable value. In practice, the electron escape depth will be less due to defects scattering, so the actual spin polarization will be larger. For the photon-energy-dependent measurements, although they are helpful to confirm the origin of the spin polarization, we would like to emphasize that our present experimental and theoretical works are adequate to proof the existence of R-2 effect in the monolayer PtSe₂. On the other hand, the practical problem is the difficulty to carry out further spin-ARPES measurements immediately due to the limited beam time. Nevertheless, these measurements are still in our future plan, and we will try our best to perform further photon-energy-dependent experiments on spin polarization.

Last, it was proposed in the paper that the existence of the new spin-layer locking as a general consequence of the common sandwich type of crystal structures in layered TMDs. If so, one may wonder whether similar R-2 effect should also be observed in bulk crystal of WSe₂ as similar argument of photoelectron depth should also be applicable. In fact, previous spin-ARPES experiments [Ref. 4] reported hidden spin polarization due to D-2 effect. It would be helpful for the authors to comment in the paper the difference between the PtSe₂ monolayer and bulk crystal of WSe₂.

Reply: We agree with the reviewer, and we have revised manuscript to make it clearer.

In summary, this paper reports the first direct observation of R-2 Rashba effect in monolayer PtSe₂, which would make a timely contribution to the field in searching for spin polarization in centrosymmetric materials. The data presented in the paper is of high quality and the data presentation is well organized. I would recommend this paper for publication in Nature Communications, provided that the authors can satisfactorily address the above issues.

Reviewer #1 (Remarks to the Author):

The authors have addressed the comment I made in the first round of reviewing.

As far as I can see, they have also addressed the comments of the other reviewers.

I believe that this manuscript should now be published in Nature Comms, without further modification. It appears that all of the reviewers are positive about the manuscript and that any previous criticisms have been addressed.

[In future, I would strongly encourage the authors to upload a manuscript in which the changes have been marked/highlighted.]

Reviewer #2 (Remarks to the Author):

All my questions were considered and properly replied. Also the questions of other referees were also correctly replied. The manuscript is good and interesting. Therefore I recommend the manuscript for publication in the current form.

Reviewer #3 (Remarks to the Author):

The rebuttal letter prepared by the authors has adequately addressed all my comments in previous report, except for the last one. I still feel that the following sentence could be a bit misleading: "The existence of the new spin-layer locking can be understood as a general consequence of the common sandwich type of crystal structures in all these materials, in which local electric fields are expected to point from the two outer Se layers to the central Pt layer." The reader may wonder whether there is anything special about PtSe₂, or such an R-2 effect should be expected in any MX₂ monolayer thin film. In either case, it would be good to for the authors to make it clear in the paper.

Other than this minor comment, I don't have any objection for publishing this paper in Nature Communications.

Reviewer #3 (Remarks to the Author):

The rebuttal letter prepared by the authors has adequately addressed all my comments in previous report, except for the last one. I still feel that the following sentence could be a bit misleading: "The existence of the new spin-layer locking can be understood as a general consequence of the common sandwich type of crystal structures in all these materials, in which local electric fields are expected to point from the two outer Se layers to the central Pt layer." The reader may wonder whether there is anything special about PtSe₂, or such an R-2 effect should be expected in any MX₂ monolayer thin film. In either case, it would be good to for the authors to make it clear in the paper.

Reply: we appreciate the pertinent suggestion from reviewer #3. Actually, our work reveals a general mechanism of R-2 effect in sandwiched crystal structure, which is less relevant to the specific elements in such a structure. We have revised our manuscript accordingly, "The existence of the new spin-layer locking can be understood as a general consequence of the common sandwich type crystal structures in all these materials, in which local electric fields are expected to point from the two outer layers to the central layer."

Other than this minor comment, I don't have any objection for publishing this paper in Nature Communications.